# Polylevolysine and Fibronectin-Loaded Nano-Hydroxyapatite/PGLA/Dextran-Based Scaffolds for Improving Bone Regeneration: A Histomorphometric in Animal Study

**DOI:** 10.3390/ijms24098137

**Published:** 2023-05-02

**Authors:** Elena Canciani, Paola Straticò, Vincenzo Varasano, Claudia Dellavia, Chiara Sciarrini, Lucio Petrizzi, Lia Rimondini, Elena M. Varoni

**Affiliations:** 1Department of Health Sciences, Università del Piemonte Orientale, 28100 Novara, Italy; elena.canciani@uniupo.it; 2Department of Veterinary Medicine, University of Teramo, 64100 Teramo, Italy; pstratico80@gmail.com (P.S.); vvarasano@unite.it (V.V.); chiarasciarrini@icloud.com (C.S.); lpetrizzi@unite.it (L.P.); 3Department of Biomedical, Surgical and Dental Sciences, Università degli Studi di Milano, 20142 Milan, Italy; claudia.dellavia@unimi.it (C.D.); elena.varoni@unimi.it (E.M.V.)

**Keywords:** bone regeneration, hydroxyapatite, fibronectin, Polylevolysine, critical-size bone defect

## Abstract

The regeneration of large bone defects is still demanding, requiring biocompatible scaffolds, with osteoconductive and osteoinductive properties. This study aimed to assess the pre-clinical efficacy of a nano-hydroxyapatite (nano-HA)/PGLA/dextran-based scaffold loaded with Polylevolysine (PLL) and fibronectin (FN), intended for bone regeneration of a critical-size tibial defect, using an ovine model. After physicochemical characterization, the scaffolds were implanted in vivo, producing two monocortical defects on both tibiae of ten adult sheep, randomly divided into two groups to be euthanized at three and six months after surgery. The proximal left and right defects were filled, respectively, with the test scaffold (nano-HA/PGLA/dextran-based scaffold loaded with PLL and FN) and the control scaffold (nano-HA/PGLA/dextran-based scaffold not loaded with PLL and FN); the distal defects were considered negative control sites, not receiving any scaffold. Histological and histomorphometric analyses were performed to quantify the bone ingrowth and residual material 3 and 6 months after surgery. In both scaffolds, the morphological analyses, at the SEM, revealed the presence of submicrometric crystals on the surfaces and within the scaffolds, while optical microscopy showed a macroscopic 3D porous architecture. XRD confirmed the presence of nano-HA with a high level of crystallinity degree. At the histological and histomorphometric evaluation, new bone formation and residual biomaterial were detectable inside the defects 3 months after intervention, without differences between the scaffolds. At 6 months, the regenerated bone was significantly higher in the defects filled with the test scaffold (loaded with PLL and FN) than in those filled with the control scaffold, while the residual material was higher in correspondence to the control scaffold. Nano-HA/PGLA/dextran-based scaffolds loaded with PLL and FN appear promising in promoting bone regeneration in critical-size defects, showing balanced regenerative and resorbable properties to support new bone deposition.

## 1. Introduction

Large bone defects, also called critical-size bone defects, caused by trauma, disease, or tumor resection still represent a global social burden, with a deep clinical and economic impact, high rates of complications, need of reoperations, and poor functional outcomes [1]. Critical-size bone defects show a low spontaneous regenerative capacity, and the current gold standard option of care is the transplantation of vascularized autologous tissue from other bone sites, such as the fibula, scapula, or iliac crest, although they are not exempted from donor site morbidities, long hospital stays, or impaired functions of damaged tissue [2]. Bone tissue engineering aims to face this challenging condition and propose therapeutic alternatives. Biodegradable scaffolds play a pivotal role in bone tissue regeneration to enhance osteogenesis, thanks to their structural and chemical properties [3,4]. Three-dimensional (3D) scaffolds maintain the space for newly deposed bone by providing a volume that the bone cells can populate and colonize, producing the extracellular matrix (ECM) [3]. The biocompatibility, mechanical properties, and biodegradability of employed biomaterials are pivotal to enhance the osteogenic process [5].

Hydroxyapatite (HA) is the major mineral component of natural bone, and it is one of the most investigated and used inorganic biomaterials for bone regeneration. HA ceramics are biocompatible, with a high osteogenic potential and low biodegradability [6,7], but also possess poor mechanical properties, including brittleness, low fatigue, and mechanical stress resistance, restricting their clinical applications [8]. In the last decades, nanostructured hydroxyapatite (nano-HA) has been proposed as an alternative, since its morphology increases the contact area with the host, resulting in a higher biodegradability and higher biological activity, able to accelerate the bone regeneration process [9,10]. Nano-HA has been reported to display a specific surface area and ultrafine structure, which are close to biological apatites and result in the appropriate adhesion and interaction of cells with surfaces [11]. Nano-HA significantly promotes the mineralization of the newly deposed bone matrix [12]; thus, it can be used alone or in composition with other biodegradable materials to improve bone healing and tissue regeneration [13]. Nano-HA can be combined with biodegradable synthetic polyesters, such as poly-lactic acid (PLA), poly-glycolic acid (PGA), and their copolymer poly-lactide-co-glycolide (PLGA), which have also been reported to support bone tissue regeneration [14]. PLGA, in particular, showed tunable degradation rates, excellent mechanical properties, biocompatibility, processability, and osteoinductivity [15,16]. The combination of HA with PLGA showed enhanced migration and adhesion of the bone marrow stromal stem cell during the bone regenerative process. Composite biomaterials based on NPs, in particular, have significant advantages over micro- or sub-micro-particulates, since they can be easily manipulated, express a closer contact with the surrounding tissue, and have a fast resorption [11].

Further components of osteogenic scaffolds are dextran and poly-ethyl-glycol (PEG), often used as excipients to improve the final product’s handling and promoting easy material’s extrusion during the application [17]. Dextran is a highly hydrosoluble polymer that can be completely degraded when the material is implanted, thus increasing the porosity of the scaffold [18]. Porosity is essential for cell homing and related bone regeneration: pores of 100–800 µm in diameter are considered appropriate for HA scaffolds to promote the differentiation of mesenchymal cells to osteoblasts [19].

To further promote bone regeneration, poly-L-lysine (PLL) and fibronectin (FN), which are constituents of the extracellular matrix (ECM), have been shown to improve osteoblast adhesion within the scaffold, showing osteoconductive properties [20,21,22,23], and to promote cell differentiation and the interaction of scaffolds with mesenchymal stem cells, resulting in an enhanced osteogenic potential [22,24]. PLL and FN represent promising molecules for tissue engineering [25,26], to be better explored in composite scaffolds intended for the bone regeneration of critical-size defects.

Therefore, this study aims at assessing the bone regenerative in vivo potential of a scaffold made of nano-HA, dextran, PLA/PLGA copolymer, and PEG absorbed with Poly-L-lysine (PLL) and fibronectin (FN), using a critical defect in the tibia of an ovine model.

## 2. Results

### 2.1. Scaffolds Physicochemical Characterization

#### 2.1.1. Structural Characterization of HA

The X-ray diffraction patterns of nano-HA displayed the typical diffraction maxima of a single hydroxyapatite phase (JCPDS 9-432) (Figure 1A). The diffraction maxima were well-defined, indicating a relatively high degree of crystallinity. The TEM images showed nano-HA crystals with plate-like morphology and with lengths ranging between 80 and 120 nm (Figure 1B).

#### 2.1.2. Morphological Characterization

Composite scaffolds were obtained by the compression method to include the bioactive nano-HA powders in the polymeric scaffold based on the biodegradable polyester PLGA and the porogenic polymer dextran. A SEM analysis of the scaffold surfaces (Figure 2A,B) and transversal and sagittal sections (Figure 2C,D, respectively) revealed the presence of submicrometric crystals on the surfaces and within the scaffolds. A light microscope analysis showed the macroscopic 3D architecture and the presence of porosity within both the scaffolds; pores were less visible in the control scaffold than in the test scaffold loaded with PLL and FN (Figure 2E,F, respectively). Considering two microscopic sections for each scaffold, the total porosity reached 0.53% for the control biomaterial compared to 4.87% for the test biomaterial. The mean pore dimension of the test scaffolds was 390 µm ± 40 µm, while, in the case of the control scaffolds, the mean pore size was 110 µm ± 50.

### 2.2. Surgery

All the sheep recovered from anesthesia without surgical complications. No adverse effects were observed after implantation or during hospitalization. On the second day after surgery, every animal was weight-bearing on the operated limb. No systemic or local disorders were identified. Skin sutures were removed two weeks after surgery.

### 2.3. Histological and Histomorphometric Analyses

For the histological and histomorphometric analyses, 80 slides were examined. The sections were observed at the morphological level, showing a normal bone structure with no inflammatory reaction detectable around the scaffolds.

The bone ingrowth (BI), residual material, and medullary space were evaluated at 3 and 6 months after surgery. New bone formation was present inside the defects of both the control scaffold and test scaffold after 3 months (Figure 3A–D) and, to a greater extent, 6 months after the intervention (Figure 4A–D). Residual biomaterial related to the scaffolds was still present in both defects (Figure 4B,D). The empty defects, used as the control and not filled with scaffolds, showed negligible bone regeneration at the periphery and close to the periosteum (Figure 5A,B).

The histomorphometric results are reported in Figure 6A,B. No significant difference between the residual biomaterials of both scaffolds was found 3 months after the surgical intervention, while, at 6 months, the percentages of still residual material and regenerated bone were significantly higher in the defects filled with the test scaffold (scaffold B) than in those with the control scaffold (scaffold A) (*p* = 0.04) (Wilcoxon paired test). No significant differences in the analyzed histomorphometric parameters were observed for both the control scaffold and for the test scaffold (*p* > 0.05) (Wilcoxon unpaired test) when comparing the two time points (3 months versus 6 months).

## 3. Discussion

The increasing request for novel biocompatible bone substitutes represents a clinical priority for regenerative medicine. The current study showed that both the control scaffold (78% of nano-HA and 28% dextran) and the test scaffold (78% of nano-HA and 28% dextran absorbed with PLL and FN) were effective in bone regeneration for the model of a circular critical-size defect on sheep tibiae. This “drill hole” critical-size defect model aims at evaluating the in vivo bone healing, avoiding the need for, and the influence of, fixation devices to stabilize the fracture [27,28,29].

In the current study, the test scaffold included FN and PLL loading. Both the control and test scaffolds showed adequate porosity and 3D architecture for promoting vascularization and new bone regeneration into the inner part of the scaffold themselves, as supported by the histological analysis [30]. The two scaffolds showed different porosities, which were higher in the test scaffold compared to the control one; this aspect could have influenced the regenerative process, promoting bone formation thanks to a better vascular and cell supply. No adverse reaction was observed during the experiment; the biomaterials and their intermediate products were confirmed safe, consistent with the previous literature [21,31,32,33]. At 6 months, the test scaffold was significantly more resorbed and replaced with regenerated bone than the control scaffold. As expected, in a critical-size defect, the empty control sites showed only a small amount of regenerated bone at the periphery of the defects and close to the periosteum.

Nano-HA was the major component of the scaffolds used in this study; it was selected because the HA nanocrystals are resorbed faster, improving the osteoconductive and mechanical properties of the scaffold [34,35]. A previous study supported the combined PLGA/nano-HA fibers in promoting osteoblast proliferation, suggesting a promising application in bone tissue engineering [36]. A further PLGA/HA/PVA scaffold, intended for cartilage regeneration, showed a mechanical behavior similar to the native cartilage and appeared to induce rabid chondrocyte adhesion and proliferation, retaining its biocompatibility in vitro [37].

Nano-HA has also been also incorporated into electrospun nanofibrous PCL scaffolds, then coated with FN and tested for bone regeneration using mouse mesenchymal stem cells (mMSCs); cell attachment, proliferation, and enhanced osteogenic differentiation were established, confirming the synergistic effect of FN and nano-HA, both in vitro and in vivo [38]. Along these lines, FN was associated with gelatin as the coating in a silk fibroin scaffold intended for bone regeneration, promoting osteogenesis in a pore size-dependent manner [39]. The presence of FN appears to promote scaffold–cell interactions, as described in a study investigating MSCs and FN-coated polycaprolactone-polyurethane (PCL-PU) scaffolds [33].

PLL can be another molecule able to promote osteogenicity. A recent study reported that a ε-poly-L-lysine (EPL)-coated nanoscale polycaprolactone/hydroxyapatite (EPL/PCL/HA) composite scaffold enhanced the antibacterial and osteogenic properties in vitro and, in a rabbit calvaria bone defect model, showed a higher bone repair capacity at both 4 and 8 weeks while reducing the immunogenic reaction [40]. Similarly, PLL has been reported to enhance cell proliferation and osteogenesis when used as a coating for scaffolds or implants [21,41].

Despite the limitations of this study, which provides preclinical evidence of bone regeneration and should be complemented with a better understanding and characterization of cell biology and cell differentiation within the 3D scaffold porous architecture, as well as a better characterization of the blood vessel supply, the PLL- and FN-loaded nano-HA/PGLA/dextran-based scaffolds appear promising in bone tissue regeneration. This study confirmed, in vivo, the role of PLL and FN in enhancing osteogenic properties.

Future perspectives may include the use of customizable vascularized transplantable bone scaffolds [2], having long-term mechanical stability and limiting the potential complications associated with the vascularized autologous bone used routinely in clinics, accelerating the patient’s recovery and improving their quality of life.

## 4. Materials and Methods

### 4.1. Materials

All the common high-purity (or cellular grade) chemical reagents were supplied by Sigma-Aldrich^®^, St. Louis, MO, USA. Poly (dl lactide-co-glycolide) RG504 was purchased from Boehringer^®^ (Ingelheim am Rhein, Germany).

### 4.2. Scaffold Fabrication

Nano-HA were prepared from an aqueous suspension of Ca(OH)_2_ (1.35 M, 1 L) by the slow addition of aqueous H_3_PO_4_ (1.26 M, 600 mL) at 95 °C at the standard atmospheric pressure. The reaction mixture was kept under stirring (always at 95 °C) for 4 h; then, the stirring was suspended, and the mixture was left standing for 2 h to allow deposition of the inorganic phase. This latter was isolated by filtration of the mother liquor, repeatedly washed with water, and dried in the oven at 60 °C for 24 h.

The control scaffold (scaffold A) was obtained by pressing at 5 tons for 10 min, dextran, nano-HA powders, and PLGA copolymer (53 wt%, 45 wt%, and 2 wt%, respectively). The test scaffold (scaffold B) was obtained by further soaking the scaffold in PLL and FN aqueous solution (0.01 wt% and 0.002 wt%, respectively) for 20 min at 37 °C.

### 4.3. Scaffold Physicochemical Characterization

#### 4.3.1. Structural Characterization of HA

Transmission electron microscopy (TEM) was carried out using a Philips CM 100 instrument (80 kV) to investigate the HA morphology. The powdered samples were ultrasonically dispersed in ultrapure water, and then, a few slurry droplets were deposited on holey carbon foils supported on conventional copper microgrids. X-Ray diffraction (XRD) powder patterns were collected using Analytical X’Pert Pro equipped with a X’Celerator detector powder diffractometer using Cu Ka radiation generated at 40 kV and 40 mA. The instrument was configured with 1/2 divergence and receiving slits. A quartz sample holder was used. The 2 ϑ range was from 5 to 60, with a step size (2 ϑ) of 0.05 and a counting time (s) of 3.

#### 4.3.2. Morphological Characterization

The morphology of the scaffolds was examined with a scanning electron microscope (SEM) (JEOL JSM 5410, JEOL Ltd., Tokyo, Japan). The samples were mounted on aluminum stubs and sputter-coated with gold.

### 4.4. In Vivo Experiments

#### 4.4.1. Animals

Ten Italian Appenninica sheep, non-pregnant females aged 2–4 years (mean 30 ± 5.2 months) and 48–64 kg body weight (mean 54.2 ± 4.8 kg), were selected for the study. All the animals were healthy and not affected by any orthopedic disorder. Therefore, the national government agency responsible for animal welfare and protection (n. 116/09) authorized the experimental protocol (ethical approval).

The animals were randomly divided into two groups: Group 1 and Group 2, scheduled to be sacrificed at different experimental times (3 months and 6 months after surgery, respectively). Before surgery, every animal received antibiotics (Oxytetracycline 11 mg/kg i.m., Panterramicina^®^-Pfizer Italia Srl Div Vet, Roma, Italy) and anti-inflammatory therapy (flunixin meglumine 1.1 mg/kg i.m., Meflosyl^®^-Fort Dodge Animal Health Spa, Bologna, Italy).

#### 4.4.2. Surgery

General anesthesia was induced with a single dose of xylazine (0.2 mg/kg i.v., Megaxilor^®^-Bio 98 Srl, Milano, Italy) and ketamine hydrochloride (6.5 mg/kg i.v., Ketavet^®^ 100-Intervet Productions Srl, Aprilia, Italy) mixed with diazepam (0.05 mg/kg i.v., Diazepam 0.5%^®^, Intervet Productions Srl, Aprilia, Italy). The sheep were intubated, and anesthesia was maintained using isoflurane in the oxygen. Once in dorsal recumbency, the medial side of both tibiae of each animal was trichotomized and surgically prepared. A 4 cm vertical incision on each tibia’s sub-condylar cranial plateau was made through the skin down to the periosteum. Once the medial cortex of the bone was exposed, two holes were drilled through the cortex; a hole was made 2 cm distal to the tibial tuberosity and a second hole 2 cm distal to the first one. Both holes measured 6 mm in diameter and 4 mm in depth [27]. All the proximal defects were filled with scaffold: scaffold A (control scaffold) in the left tibias and scaffold B (test scaffold) in the right tibias. The distal defects were left empty and were used as negative controls (empty defect control). One centimeter caudal to each defect, a 2 mm threaded pin was placed through the cortex as a marker, cut at the level of the outer surface of the bone. In the end, the periosteum was repositioned, the subcutis sutured with 2/0 absorbable material (Byosin^®^, Syneture, Norwalk, CT, USA) in a simple continuous pattern, and the skin with 2/0 non-absorbable material (Novafil^®^, Syneture) in a simple interrupted pattern. After surgery, all the sheep received a three-day course of flunixin meglumine (1.1 mg/kg i.m., SID-Meflosyl^®^, Fort Dodge Animal Health Spa, Bologna, Italy).

#### 4.4.3. Postoperative Histological Evaluation

Three (Group 1) and six months (Group 2) after the surgery, the animals were euthanatized with an overdose of thiopental and embutramide (Tanax^®^, Intervet Italia Srl, Milano, Italy; pentothal sodium^®^, Intervet Productions Srl, Aprilia, Italy) to collect the proximal epiphysis of each tibia for the histological and histomorphometric analyses.

The samples were immediately stored in a 4% buffered formalin at 4 °C until further processing as non-decalcified bone specimens. The specimens were dehydrated with an increasing alcohol scale infiltrated and embedded in poly-methylmethacrylate resin (Kulzer Technovit 7200 VLC^®^, Bio Optica, Milano, Italy). Each bone sample was cut longitudinally to the drill hole axis with a diamond blade (Micromet Remet^®^, Bologna, Italy) to obtain 4 sections for each block section. The slices were processed to reach 80.00 ± 20.00 µm thickness and subsequently stained with Toluidine Blue/Pyronin Y (Sigma-Aldrich^®^, St. Louis, MO, USA). They were examined at different magnifications (2.5×, 10×, 20×, and 40×) with an optical microscope (Eclipse E600^®^) and stereomicroscope (SMZ800^®^, Nikon, Tokyo, Japan) connected to a digital camera (DXM1200^®^, Nikon, Tokyo, Japan). Bone ingrowth (BI), residual material, and medullary space were calculated using image analysis software “Image J” (NIH^®^, Bethesda, MD, USA) [42,43].

### 4.5. Statistics

Data obtained from the histological analysis were compared at each time point (T3 and T6), and the histomorphometric data (percentage of medullary space, residual material, and bone ingrowth) were compared between the two different scaffolds (control scaffold A versus test scaffold B) using the Wilcoxon signed-rank test for paired data. In addition, the histomorphometric data obtained in defects filled with scaffold A (control) or with scaffold B (test) were compared between the two time points at 3 and 6 months, using the Wilcoxon rank-sum test for unpaired data. The level of significance (*p*) was set at 5%.

The mean and standard deviation for each parameter and time point were computed separately for scaffolds A (control scaffold) and B (test scaffold).

## 5. Conclusions

Scaffolds based on nano-HA/PLGA resulted in new bone formation in an ovine model system of critical-size cortical defects. Evidence of new bone formation was detectable inside the defects in both the control scaffold and the test scaffold after 3 months. The presence of PLL and FN resulted in scaffolds with higher porosity and enhanced bone regeneration; at 6 months, the scaffolds loaded with PLL and FN were significantly more resorbed and replaced with regenerated bone compared to the control scaffold, while the empty sites, used as controls, showed only a small amount of regenerated bone at the periphery of the defects and close to the periosteum. PLL and FN loading considerably improved the osteoconductive properties, supporting this strategy of biomaterial functionalization as promising in bone tissue engineering.

## Figures and Tables

**Figure 1 ijms-24-08137-f001:**
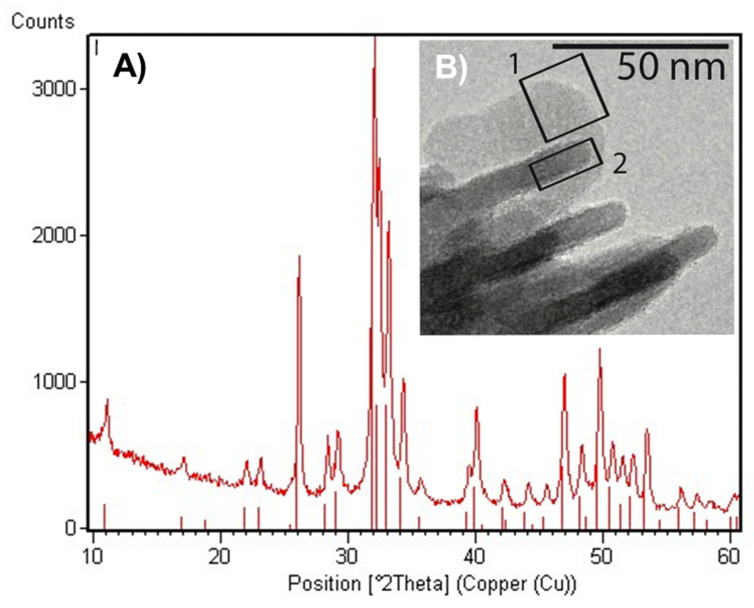
Structural characterization of HA. (**A**) XRD diffraction patterns displayed the typical diffraction maxima of a well-defined single hydroxyapatite phase, thus suggesting a high crystallinity degree; (**B**) TEM image showed well-defined nano-HA crystals differently oriented with respect to the image plane (indicated in squares 1 and 2, two different projections), confirming the high crystallinity degree.

**Figure 2 ijms-24-08137-f002:**
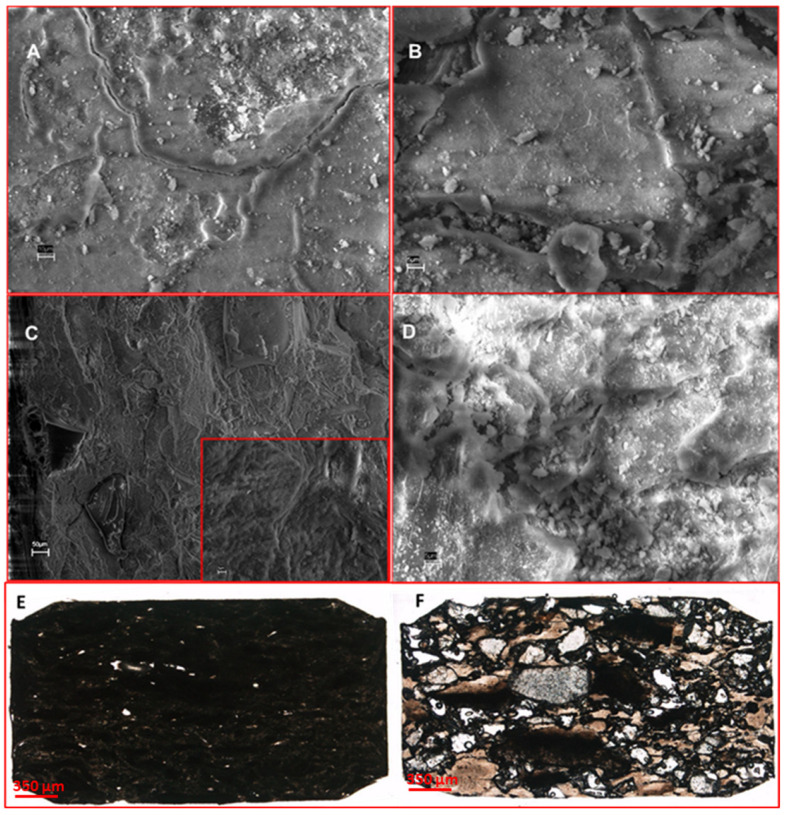
Morphological analyses of the scaffolds. The SEM analysis of the scaffold surfaces (**A**,**B**, scale bar = 10 µm and 5 µm) showed the presence of submicrometric crystals, also highlighted at the transversal section (**C** and insert, scale bar = 50 µm and 5 µm, respectively) and sagittal sections (**D**, scale bar = 5 µm). In addition, the ground sections of the control scaffold (**E**) and test scaffold (**F**) with the light microscope show the presence of porosity within both the scaffolds; the control scaffold was more compact, with less porosity, than the test scaffold (total magnification 4×).

**Figure 3 ijms-24-08137-f003:**
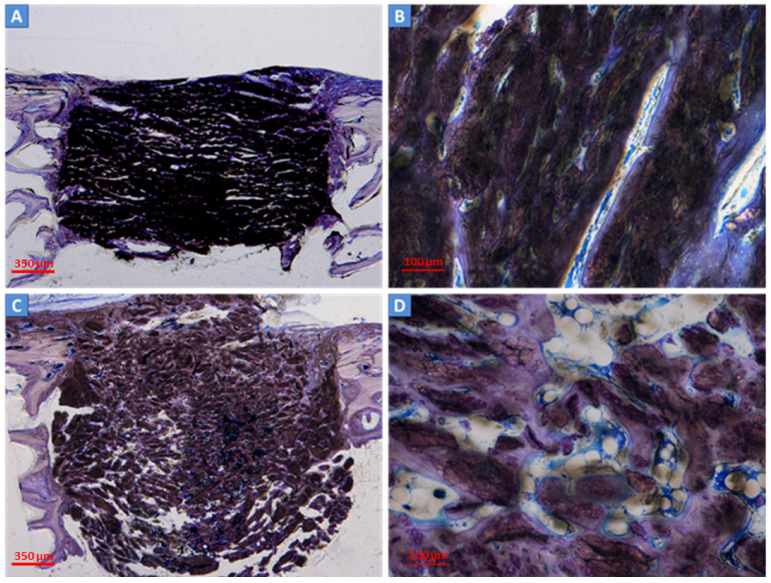
Histological analysis of the scaffolds at 3 months after surgery. The new bone deposition was observed within the control scaffold (**A**,**B**) and test scaffold (**B**,**D**) at three months. Deposited bone (in blue/light violet) was detectable at the periphery (**A**,**C**) and inside the defect (**B**,**D**). The residual material was still present (in dark brown). Light microscope, total magnification 4× (2A, 2C) and 10× (2B, 2D); Toluidine Blue/Pyronin Yellow staining.

**Figure 4 ijms-24-08137-f004:**
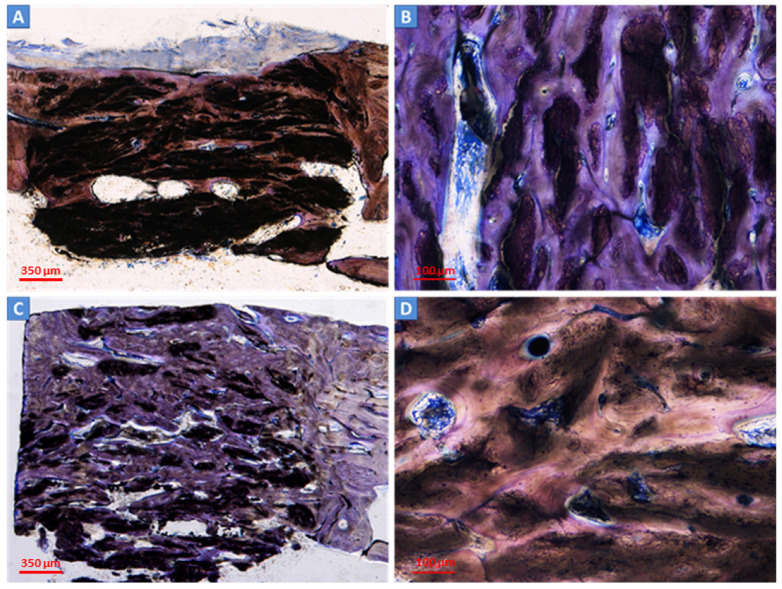
Histological analysis of the scaffolds at 6 months after surgery. At six months, a high amount of new bone (in blue/light violet) was found in the defects filled with the control scaffold (**A**,**B**) and test scaffold (**B**,**D**). The residual material was still present in both scaffolds (in dark brown). Light microscope, total magnification 4× (3A, 3C) and 10× (3B, 3D); Toluidine Blue/Pyronin Yellow staining.

**Figure 5 ijms-24-08137-f005:**
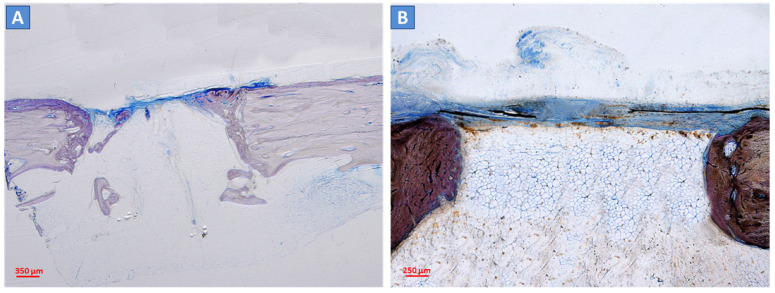
Histological analysis of empty defects, used as the control, at 3 and 6 months after surgery. The empty defects showed a negligible amount of new bone detectable only at the periphery of the defect at both 3 and 6 months after surgical intervention (**A**,**B**, respectively). Light microscope, total magnification 4×; Toluidine Blue/Pyronin Yellow staining.

**Figure 6 ijms-24-08137-f006:**
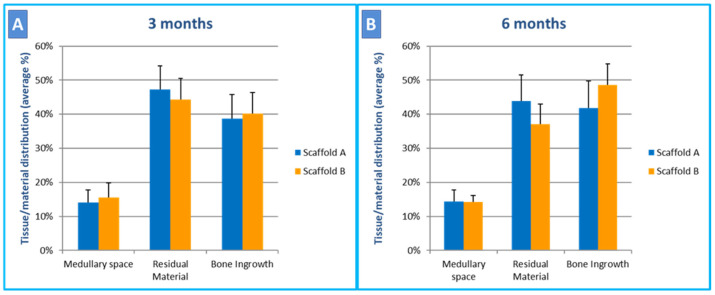
Histomorphometric analysis of new bone formation. (**A**) At 3 months, residual intra-defects biomaterial and bone ingrowth were comparable between the scaffolds. (**B**) At 6 months, scaffold B (test scaffold) was significantly more resorbed compared to scaffold A (control scaffold) and showed a significantly increased bone ingrowth (4% for control scaffold A and 7% for test scaffold B).

## Data Availability

The data presented in this study are available on request from the corresponding author.

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
