# Peer review of "Polylevolysine and Fibronectin-Loaded Nano-Hydroxyapatite/PGLA/Dextran-Based Scaffolds for Improving Bone Regeneration: A Histomorphometric in Animal Study"

_ijms, 2023, doi:10.3390/ijms24098137_

Round 1

Reviewer 1 Report

In this study, the authors aimed to evaluate the pre-clinical efficacy of a nano-hydroxyapatite (nano-HA)/PGLA/dextran-based scaffold loaded with Polylevolysine (PLL) and fibronectin (FN) in addressing critical-size tibial defects. The results demonstrated that nano-HA/PGLA/dextran-based scaffolds loaded with PLL and FN effectively promoted bone regeneration. However, I have some concerns regarding the experimental design and results, particularly animal number, and I’d like to recommend a major revision prior to acceptance.

 1. Please specify the number of replicates for each group in the animal study. Is the current sample size or animal population sufficient to support the claims made in the study?

2. The abstract mentions the mechanical properties of the newly regenerated bones. Where are the corresponding results presented in the paper?

3. Figures 2E and 2F depict scaffolds with notably different porosities. Given that porosity can significantly impact regeneration outcomes, could such a substantial difference be the primary factor influencing the in vivo results?

Author Response

In this study, the authors aimed to evaluate the pre-clinical efficacy of a nano-hydroxyapatite (nano-HA)/PGLA/dextran-based scaffold loaded with Polylevolysine (PLL) and fibronectin (FN) in addressing critical-size tibial defects. The results demonstrated that nano-HA/PGLA/dextran-based scaffolds loaded with PLL and FN effectively promoted bone regeneration. However, I have some concerns regarding the experimental design and results, particularly animal number, and I’d like to recommend a major revision prior to acceptance.

R: We are grateful for the useful comments which allowed us to better explain the experimental design and results.

  1. Please specify the number of replicates for each group in the animal study. Is the current sample size or animal population sufficient to support the claims made in the study?

R: Thank you for the comment. We performed two defects in each sheep tibia; each tibial bone had a test site (receiving the scaffold) and a control site (empty). A total of 10 animals were used. Sample size calculation allowed us to achieve the 60% of statistical power, in terms of regenerated bone after 6 months, able to support the findings of the study. Due to ethical reasons and considering that this was the first animal study evaluating the proposed scaffolds, we calculated the number of animals basing on previous literature, on expected results and on the ethical reasons related to the need to avoid killing animals unnecessarily at the end of the study.

  1. The abstract mentions the mechanical properties of the newly regenerated bones. Where are the corresponding results presented in the paper?

R: We apologize for the inaccuracy. We did not test directly the mechanical properties of the scaffolds, but we deduced the potential in terms of mechanical resistance to sustain the weight by observing the animals used in the study, which demonstrated that all the animal recovered completely from the intervention and, on the second day after surgery, every animal was weight bearing on the operated limb. However, we agree that the sentence is inaccurate and we corrected the abstract accordingly, by removing the mention to mechanical properties.

  1. Figures 2E and 2F depict scaffolds with notably different porosities. Given that porosity can significantly impact regeneration outcomes, could such a substantial difference be the primary factor influencing the in vivo results?

R: Thanks for the important comment. We agree that the differential porosities between the two scaffolds may have influenced the bone regeneration. We added a specific sentence on this in the Discussion.

“The two scaffolds showed different porosities, which were higher in the test scaffold compared to the control one: this aspect could have influenced the regenerative process, promoting the bone formation thanks to a better vascular and cell supply.”

Reviewer 2 Report

Originality of this paper is well explained and research design is basically good. This is acceptable after the following minor corrections.

(1) Meaning of square 1 and 2 in Fig. 1 is unclear.

(2) Chemical formula should be subscript in Line 225 and 226.

Author Response

Originality of this paper is well explained and research design is basically good. This is acceptable after the following minor corrections.

R: Many thanks for the kind feedback.

(1) Meaning of square 1 and 2 in Fig. 1 is unclear.

R: We apologize for the inaccuracy. The number 1 and 2 within the square referring to the TEM image highlights nano-HA crystals in different projections. We changed the caption indicating the square 1 and 2 as reported below:

“(B) TEM image showed nano-HA crystals differently oriented with respect to the image plane (indicated in squares 1 and 2, two different projections), well-defined, confirming the high crystallinity degree”

(2) Chemical formula should be subscript in Line 225 and 226.

R: Sorry for the mistake. Corrected as kindly recommended.

Reviewer 3 Report

Manuscript “Polylevolysine and fibronectin-loaded nano-hydroxyapatite/PGLA/dextran-based scaffolds for improving bone regeneration: a histomorphometric in animal study” represents a contribution to field of molecular sciences. Text is clear and easy to read. The research topic is original. The submitted manuscript in the subject area, compared to other published material, is a contribution especially because it has in vivo studies. All references used are appropriate.

Before accepting the manuscript, it is essential that the authors:

·       It is necessary to emphasize in the introduction why nano particles of hydroxyapatite  have an advantage over micro particles, and in combination with bioresorbable polymers: https://doi.org/10.1002/jbm.b.31630 (include reference)

·       In sistem polylevolysine and fibronectin-loaded nano-hydroxyapatite/PGLA/dextran-based scaffolds, it is necessary to determine and display the results of loading efficiency – quantitatively (for polylevolysine and fibronectin).

·       It is necessary to supplement the results, especially the discussion, with values of total porosity of scaffolds.

·       It is necessary to define the pore size of the polylevolysine and fibronectin-loaded nano-hydroxyapatite/PGLA/dextran-based scaffolds.

·       The conclusion is too general. It needs to be improved.

Author Response

Manuscript “Polylevolysine and fibronectin-loaded nano-hydroxyapatite/PGLA/dextran-based scaffolds for improving bone regeneration: a histomorphometric in animal study” represents a contribution to field of molecular sciences. Text is clear and easy to read. The research topic is original. The submitted manuscript in the subject area, compared to other published material, is a contribution especially because it has in vivo studies. All references used are appropriate.

R: We are grateful for the overall positive feedback on our manuscript and thanks for the suggestions, which contributed to implement the paper.

Before accepting the manuscript, it is essential that the authors:

  • It is necessary to emphasize in the introduction why nano particles of hydroxyapatite have an advantage over micro particles, and in combination with bioresorbable polymers: https://doi.org/10.1002/jbm.b.31630 (include reference)

R: We are grateful for the useful suggestion. We added in the introduction a sentence on the advantages of nano-HA over micro-particles and in combination with bioresorbable polymers. We added the suggested reference.

“Nano-HA has been reported to display a specific surface area and ultrafine structure, which are close to biological apatites and result in appropriate adhesion and interaction of cells with surfaces (11). […] The combination of HA with PLGA showed enhanced migration and adhesion of the bone marrow stromal stem cell during bone regenerative process. Composite biomaterials based on NPs, in particular, have significant advantages over micro- or submicro-particulates, since they can be easily manipulated, express a closer contact with the surrounding tissue, a fast resorption (11).”

  • In system polylevolysine and fibronectin-loaded nano-hydroxyapatite/PGLA/dextran-based scaffolds, it is necessary to determine and display the results of loading efficiency – quantitatively (for polylevolysine and fibronectin).

R: Thank you for the comment. We decided the loading conditions based on the previous literature. We summarize here the main papers related to polylevolysine and fibronectin coating of biomaterials for biomedical purposes:

https://pubs.rsc.org/en/content/articlelanding/2019/nj/c9nj01675a

https://www.sciencedirect.com/science/article/pii/S0928493117342613

https://www.sciencedirect.com/science/article/pii/S0927776509001696?via%3Dihub

https://www.ncbi.nlm.nih.gov/pmc/articles/PMC4048976/

https://www.sciencedirect.com/science/article/pii/S0928493119335064

  • It is necessary to supplement the results, especially the discussion, with values of total porosity of scaffolds.

R: We are grateful for the useful comment. The total porosity was 0.53% for the control scaffold, compared to 4.87% for the test scaffold. The porosity has been calculated, using microscopy images, in 2 sections for each kind of biomaterial. We added this information within the Results.

“The total porosity resulted 0.53% for the control biomaterial, compared to 4.87% for the test biomaterial.”

  • It is necessary to define the pore size of the polylevolysine and fibronectin-loaded nano-hydroxyapatite/PGLA/dextran-based scaffolds.

R: The mean pore dimension of the test scaffolds was calculated on two histological section and resulted equal to 390 µm ± 40 µm. In case of the control scaffolds, the mean pore size was 110 µm ± 50. We added this information within the Results.

“The mean pore dimension of the test scaffolds was 390 µm ± 40 µm, while, in case of the control scaffolds, the mean pore size resulted 110 µm ± 50.”

  • The conclusion is too general. It needs to be improved.

R: Thank you for the suggestion. We changed the conclusion focusing on the most relevant findings coming from the study.

“Scaffolds based on nano-HA/PLGA resulted in new bone formation in an ovine model system of critical size cortical defect. Evidence of new bone formation was detectable inside the defects both in correspondence of the control scaffold and of the test scaf-fold after 3 months. The presence of PLL and FN resulted in scaffolds with higher po-rosity and enhanced bone regeneration: at 6 months, the scaffold loaded with PLL and FN was significantly more resorbed and replaced with regenerated bone than control scaffold, while the empty sites, used as controls, showed only a small amount of regenerated bone at the periphery of the defects and close to the periosteum. PLL and FN loading considerably improve the osteoconductive properties, supporting this strategy of biomaterial functionalization as promising in bone tissue engineering.”

Round 2

Reviewer 1 Report

The authors have addressed my concerns.